# The Effect of Increasing Neutral Detergent Fiber Level through Different Fiber Feed Ingredients throughout the Gestation of Sows

**DOI:** 10.3390/ani11020415

**Published:** 2021-02-06

**Authors:** Baoming Shi, Wei He, Ge Su, Xiaodong Xu, Anshan Shan

**Affiliations:** Institute of Animal Nutrition, Northeast Agricultural University, Harbin 150030, China; hewei951011@163.com (W.H.); suge@muyuanfoods.com (G.S.); yaxmxxd@163.com (X.X.); asshan@neau.edu.cn (A.S.)

**Keywords:** dietary fiber, sows, performance, milk composition, plasma biochemical factors, digestibility

## Abstract

**Simple Summary:**

To reduce feed costs, the exploration of feed resources is currently the main research direction. In the past, fiber feed was generally regarded as anti-nutritional factors. Recently, fiber has received special attention due to its many beneficial effects. Therefore, this study selected five common household and production wastes as fiber sources and added them to the diet of pregnant sows to explore the impact on sows and piglets. The results of this study indicated that adding corn gluten feed (CG) significantly improved the digestibility of sows and body weight (BW) of piglets, which provide evidence and insight for the application of corn gluten feed in gestation sow diets.

**Abstract:**

The purpose of this study was to determine the effect of increasing dietary neutral detergent fiber (NDF) levels on pregnant sows, and to select the best feed ingredients based on reproductive performance, plasma biochemistry parameters, colostrum and milk composition, and nutrient digestibility. Seventy-two multiparous sows were randomly allotted to six dietary treatment groups (*n* = 12). The feeding of chicory meal (CM), wheat bran (WB), corn gluten, and rice bran meal (RBM) increased the average weaning weight of piglets compared with the control (CON) group (*p* < 0.05). Supplementation with CG diet increased the sow BW, weight gain, and back fat thickness compared with WB and RBM on day 107 of gestation (*p* < 0.05). Furthermore, Supplementation with CG diet resulted in lower plasma urea nitrogen (PUN) and higher total protein (TP) concentrations in plasma (*p* < 0.05). Feeding CM diet and soybean curd residue (SCR) diet reduced the total protein and globulin, and supplementation with CM diet significantly increased the PUN (*p* < 0.05). The apparent total tract digestibility (ATTD) of crude protein (CP), crude fat (EE), calcium (Ca), phosphorus (P), neutral detergent fiber (NDF), and acid detergent fiber (ADF) were decreased following the addition of CM, WB, or SCR to the diets (*p* < 0.05). The ATTD of NDF and ADF were significantly increased in the CG group (*p* < 0.05). In conclusion, the feeding of CG diet to sows have an excellent effect.

## 1. Introduction

Diets with high neutral detergent fiber (NDF) levels are rarely used in growing and finishing pigs, which may reduce dietary energy density and digestibility. However, NDF contributes to intestinal peristalsis and maintenance of body health, which may be needed by sows [1]. According to animal welfare legislation in the European Union (council directive 2001/88/EC), all pregnant, dry sows and gilts have to be provided with sufficient quantities of bulky or high-fiber feed to alleviate feeding frustration during gestation and to prepare females for ad libitum feed supply during lactation [2]. The influence of high NDF level diets on sow has been increasingly addressed. NDF is considered to be a dietary ingredient resistant to mammalian enzyme degradation and cannot be hydrolyzed and absorbed in the small intestine [3]. However, some studies have shown that almost 20% of the NDF consumed by animals in their diet is digested before reaching the end of the ileum, which may be accomplished by bacterial fermentation [4]. In 2001, the American Association of Cereal Chemists defined fiber as edible or similar carbohydrates in plants, which cannot be digested and absorbed by the human small intestine but can be fully or partially fermented in the hindgut. Therefore, the NDF in the feed ingredients can improve the growth performance of pigs through the degradation of fiber by the intestinal flora in the hindgut and promote the improvement of the intestinal flora and fermentation indicators [5,6,7,8]. After the piglets are born, the bacteria in the intestine and milk of the sow are an important source of intestinal flora of the new born piglets [9]. Therefore, improving the diet of sows during gestation to change the composition of maternal intestines and milk flora, thereby improving the immunity, disease resistance and survival rate of piglets, is the key to improving production performance through the integration of sows and sons.

Prenatal and early postnatal life is a critical period for the development of piglet [10], Many studies have shown that malnutrition in sows during gestation can adversely affect the growth and health of offspring [11]. Therefore, maternal nutrition during sow gestation has a crucial role in piglet growth and development [12]. A proportion of NDF ranging from 15 to 20% enables sows to adjust their daily feed intake and caters to their needs [13]. At present, most of the studies on fiber feed for sows are focused on controlling the level of dietary fiber, the addition of fiber to gestation diets does not produce better results for the reproductive performance of sows [14,15]. In contrast, feeding a high-fiber diets during gestation can increase voluntary feed intake during lactation and improve sow reproductive performance [16]. However, there are few studies to evaluate the effect of dietary control in terms of NDF level. In the past, it was believed that fiber had an anti-nutritional effect on pigs [17,18]. However, with the in-depth research on fiber nutrition, it was found that improving the level of fiber or improving the source of fiber showed good results in sow breeding process [2,19]. The exploration of fiber feed resources has gradually become a research hotspot in the industry. In recent years, there have been many disputes about the application of NDF in sow diets. Therefore, it is necessary to conduct research to evaluate the effect of dietary NDF on sows and piglets.

The diversity of sources and chemical structures determines the unique physical and chemical properties of dietary fiber, including four aspects of cation exchange capacity, hydration characteristics, viscosity and fermentability [20]. The addition of different types or different amounts of fiber sources to the diets of pregnant sows produced different effects. This study explored the effects of including wheat bran (WB), chicory meal (CM), soybean curd residue (SCR), corn gluten (CG) and rice bran meal (RBM) diets to the sow gestation, these feed ingredients are rich in NDF, and control the experimental diets contained 16% NDF. The present study aimed to determine the effects of different fiber-derived diets on sow and piglet performance, blood biochemistry, nutrient digestibility and milk composition compared with those of corn-soybean meal diets and provide options for finding suitable fiber materials for sow of gestation.

## 2. Materials and Methods

The protocols used in this experiment were approved by the Northeast Agricultural University Institutional Animal Care and Use Committee of China (NEAU-(2011)-9). 

### 2.1. Animals and Experimental Design

A total of seventy-two sows (Large White × Landrace) with 3 to 4 parity were used in this trial, and the number of sows at 3 and 4 parities was thirty-six, respectively. They were inseminated artificially three times and randomly assigned to six diet treatment groups with the similar body condition between treatment groups. Each treatment group was composed of 12 sows, of which 3 parities and 4 parities are half and half, respectively. The dietary treatment groups included (1) a corn-soybean meal basal diet (CON, *n* = 12), (2) basal diet with 36.4% wheat bran (WB, *n* = 12), (3) basal diet with 23.8% chicory meal (CM, *n* = 12), (4) basal diet with 17.6% soybean curd residue (SCR, *n* = 12), (5) basal diet with 27% corn gluten (CG, *n* = 12), and (6) basal diet with 46.5% rice bran meal (RBM, *n* = 12). The CON diet contained 8.4% NDF, and the experimental diets, which contained 16% NDF, were fed to sows from day 1 to day 107 during gestation. Sows were freely fed the CON diet from day 108 until weaning (day 21) during lactation. Sows were fed according to their daily nutrition requirements, which were based on the criterion of the NRC 2012 model [21]. The ingredients are shown in Table 1.

### 2.2. Housing, Feeding, and Management

The diet was provided for sows twice a day. The sows were checked daily to judge whether they had returned to estrus using a mature boar. Pregnant sows were identified ultrasonically on the 20th day after insemination. Sows were eliminated from the experiment if they were not pregnant. Finally, a total of 72 pregnant sows were used. These pregnant sows were randomly divided into six groups corresponding to the six diets. Sows were rebred under the same conditions. Afterwards, the sows were transferred to the farrowing house. The farrowing room was environmentally controlled, and the average ambient daily temperature was approximately 18–20 °C. The sows were fed 1.8 kg/day of the gestation diet from day 1 to 30, 2.4 kg/d from day 31 to 90, and 3.0 kg/d from day 91 to the day of the expected date of confinement. On the day of farrowing, the sows were not fed. The feed allowance of the sows was raised to 3.0 kg on day 1. Thereafter, this amount was increased daily by 1.0 kg until the maximum was reached. The sows were fed three times a day, and the feed refusals were weighed daily; the actual feed intake for each sow was recorded.

On the third day after birth, the piglets received an iron injection (Iron Dextran, Jiangxi Chuangdao Animal Health Co., Ltd., Nanchang, China). Commercial creep feed (15.8 MJ of metabolizable energy/kg, 210.0 g CP/kg, and 15.6 g lysine/kg) was offered to the piglets 7 days after birth. The intake of the creep feed was not recorded.

### 2.3. Diet Collection and Analyses

Samples of the feed were obtained from each dietary treatment. The diets were analyzed for crude protein (CP) (Kjeldahl method), NDF (Van Soest method), acid detergent fiber (ADF) (Van Soest method), crude fiber (CF) (Van Soest method), crude fat (EE) (Soxhlet extraction), Ca (titration method), P (colorimetry method) [22]. The composition analysis of fiber sources in the diet are shown in Table 2.

### 2.4. Sow and Litter Performance

The sows were weighed three times: on the day of insemination, on day 107 of gestation and on day 21 of lactation (weaning). Within the first 12 h after birth, the numbers of total piglets, born alive and dead, were recorded, and the piglets were tattooed for identification. The piglets were individually weighed at farrowing and weaning. The number of piglets that died during lactation was recorded. The survival rate at birth per treatment was analyzed after calculating the survival rate at birth per litter. The survival rate at weaning per treatment was analyzed after calculating the survival rate at weaning per litter.

### 2.5. Milk Collection and Analyses

Milk was collected on days 0 and 14 from each of the sows post-farrowing. Milk (30 mL) was collected with a 50 mL threaded pipe and hands from all of the functional mammary glands of the sows. Colostrum was collected within 24 h of when the piglets were born. Milk was collected 1 h before the sows were fed, and the sows were injected with 1 mL of oxytocin to stimulate milk release before collection on day 14 of parturition. Milk was divided into vacutainer tubes that were frozen at −20 °C until analysis.

The colostrum and milk samples were analyzed for lactose, protein, fat, and total solids with a fully automatic milk analyzer (Milko ScanTM FT + Analyzer, Foss).

### 2.6. Fecal Samples

Fecal samples were collected one time daily from day 100 to 103 of gestation and day 14 to 16 of lactation to determine the fecal nutrient digestibility in 12 sows per treatment group. Individual fecal samples from each sow at these days were mixed together, homogenized with 10 mL HCl (purity of 10%) per 100 g fecal sample, and then stored at −20 °C, dried at 60 °C and analyzed. Fecal dried samples were frozen at −20 °C until analysis.

The fecal samples were analyzed for NDF, ADF, EE, Ca, and P to calculate the apparent total tract digestibility (ATTD) of the NDF, ADF, EE, Ca, and P. The ATTD of the nutrients was calculated according to the methods of Gerritsen et al. [23].

### 2.7. Blood Sample Collection and Analyses

Heparin tubes were used to obtain blood samples (10 mL) from an ear vein of 12 sows per treatment on day 100 of gestation and day 14 of lactation. After the blood samples were centrifuged at 3000× *g* for 10 min, the plasma was partially transferred to a 1.5-mL Eppendorf (EP) tube and immediately stored at −20 °C until analysis.

The plasma samples were analyzed for plasma urea nitrogen (PUN), total protein (TP), albumin (ALB), glucose (GLU), triglyceride (TG), alkaline phosphatase (AKP), aspartate aminotransferase (AST), and alanine aminotransferase (ALT) with the UnicelDxC 800 Synchron^®^ (Clinical System, Beckman Coulter Inc., Brea, CA, USA) [24].

### 2.8. Statistical Analyses

Statistical analyses were performed using SPSS 20.0 (IBM-SPSS Inc., Chicago, IL, USA). Data was subject to a variance equality test of Levenne method. Then, one-way analysis of variance (ANOVA) was performed, and multiple comparisons were carried out using the Tukey HSD. Each pig was considered to be a statistical unit, results are presented as mean values. The data were expressed as the means ± SD (standard deviation), and a value of *p* < 0.05 was considered statistically significant. The null hypothesis of the experiment is that the diets of the CON, WB, CM, SCR, CG, and RBM groups have similar effects on pregnant sows and offspring, and the alternative hypothesis was that dietary effects of CON, WB, CM, SCR, CG, and RBM groups on pregnant sows and their offspring were different. When *p* < 0.05, the null hypothesis was rejected, when *p* > 0.05, the null hypothesis could fail to be rejected.

## 3. Results

### 3.1. Sow Performance

As shown in Table 3, compared with WB, SCR and RBM groups, treatment with CG group significantly increased sow BW on day 107 of gestation (*p* < 0.05), and no significant difference compared with the CON and CM groups was observed (*p* > 0.05). The CG diet supplement increased weight gain and back fat thickness compared with WB and RBM on day 107 of gestation (*p* < 0.05), with no significant differences among the CON, CM, and SCR groups (*p* > 0.05). Furthermore, the group fed with WB of sow BW on day 21 of weaning, sow gestation BW change, back fat thickness on day 107 of gestation and back fat gain in gestation was lower compared with the CON groups (*p* < 0.05). Changes in sow weight and back fat thickness during lactation did not differ among the treatments (*p* > 0.05).

### 3.2. Litter Performance

As shown in Table 4, Feeding CM, WB, CG, and RBM diets during gestation increased the piglet average weaning weight compared with the CON groups (*p* < 0.05), but no difference was observed in litter performance among the treatments (*p* > 0.05), including the total piglets born, piglets born alive, survival rate at birth, piglets at weaning, survival rate at weaning, average birth weight, and average daily gain (ADG).

### 3.3. Serum Biochemical Indexes

As shown in Table 5, compared with those in the CON, CM, SCR, and WB groups, sows supplemented with CG diet had a lower PUN concentration in plasma on day 100 of gestation (*p* < 0.05). In addition, supplementation with CM diet produced significantly higher concentrations of PUN in plasma compared with those in the CON, CG, and RBM groups (*p* < 0.05). The concentrations of TP in plasma on day 107 of gestation were significantly increased by the treatment supplemented with CG diet (*p* < 0.01) compared with those in the CON, CM, and SCR diets. Feeding CM and SCR diets resulted in a lower TP and globulin concentration in plasma compared with the CON groups (*p* < 0.01). Sows that were fed WB diet had higher concentrations of globulin on day 100 of gestation compared with the sows in the CON, CM, and SCR groups (*p* < 0.01). Supplementation with CG and WB resulted in higher concentrations of TP in plasma than that with SCR on day 14 of lactation (*p* < 0.01). Furthermore, supplementation with CM and SCR diets led to lower concentrations of globulin than in those in the CON diet, WB, and CG groups (*p* < 0.01). In addition, no difference was observed in the concentrations of GLU, TG, ALT, AST, AKP, and ALB during gestation and the concentrations of GLU, PUN, TG, ALT, AST, AKP, and ALB during lactation among the treatments (*p* > 0.05).

### 3.4. Composition of Colostrum and Milk

The effects of diet on the colostrum and milk of sows are shown in Table 6. There were no prominent changes in the colostrum and milk composition of the sows (*p* > 0.05).

### 3.5. Diet Nutrient Digestibility

As shown in Table 7, the ATTD of CP was increased when CON, SCR, and CG was added to the diets compared with that in the CM, WB, and RBM groups during lactation (*p* < 0.01). In addition, the ATTD of EE, Ca, and P was increased when CON and SCR was added to the diets compared with the other groups (*p* < 0.01). Moreover, the ATTD of the CM group EE and RBM group CP, Ca was lower than the other groups (*p* < 0.01). The ATTD of NDF, ADF, and CF was increased (*p* < 0.01) in the CG groups compared with CON groups (*p* < 0.01). For NDF and CF, the sows that were fed CM and RBM diets had lower ATTD compared with CON group during lactation (*p* < 0.01), and supplementation with CM diet significantly decreased the ATTD of ADF compared with the CON group (*p* < 0.01).

## 4. Discussion

### 4.1. Sow Performance

The study by Peet-Schwering et al. [25] showed that the increase in BW and back fat during gestation in the high-fiber diet group was significantly lower than in the control group, and weight and back fat loss during lactation were increased compared to that in the control group, which is consistent with the results of this experiment. High-fiber diets provide less net energy than control diets, which encourages sows to accumulate less back fat during gestation. The addition of soybean husks to the diet to increase crude fiber levels results in lower average back fat thickness [26]. In an analysis using inulin as the source of crude fiber, there was no difference in the weight of sows between the test group and control group, and the addition of inulin significantly reduced back fat thickness [27]. Our research results show that supplemented with CM and CG diet have a certain inhibitory effect on sow weight loss during gestation and back fat loss during lactation compared with supplemented high-fiber diets. Studies have shown that sow back fat thickness could reflect nutrient retention and potential reproductive efficiency [28]. However, the degree of fiber utilization by sows is also related to the physical and chemical properties of dietary fiber, the degree of lignification, solubility, time through the intestine, and degree of fermentation, among which solubility is an important factor affecting the utilization of dietary fiber in monogastric animals. The physical and chemical properties of dietary fiber will change the viscosity of intestinal chyme, the speed of intestinal passage, and the absorption of nutrients, thereby reducing the efficiency of nutrient absorption [29]. This may provide an explanation for the lower performance of sows in the WB and RBM groups.

### 4.2. Litter Performance

No significant differences were observed in the total piglets born, piglets born alive and weaning, survival rate at birth and weaning, average birth weight and ADG of piglets. The results of this experiment suggested that there were no negative effects of CM, SCR, WB, CG, and RBM diets supplementation during gestation on litter performance. Quesnel et al. [2] reported that sows fed with different fiber levels, while maintaining the same nutritional level, showed no difference in piglet birth weight. In other studies, high fiber intake during gestation had no effect on the number of live births and birth weight [19,30]. Under the same conditions of dietary energy intake, increasing the level of dietary fiber during gestation did not improve sow litter performance [31,32,33]. The above results are similar to our findings. Supplementation of high-fiber diets in the sow during gestation does not affect litter performance.

Some studies found that sows fed high-fiber diets during gestation had an increase average weaning weight of piglets [34,35]. This result is consistent with our experimental findings, in which fiber supplementation during gestation significantly increased the average weaned piglet weight compared with the CON group. In sow studies, prolactin is a key hormone for the initiation and maintenance of milk production, and a high-fiber intake tends to increase prolactin concentrations during gestation [36]. Studies have shown that diets with high fiber levels have been very successfully used to regulate sow milk production [37,38,39]. Collectively, data from the above references suggest that the level and source of fiber affect the prolactin response in pregnant sows. Because the growth of the suckling pigs was almost entirely dependent on the milk of the sows, in this study, the greater average weaning weight gain of piglets may be related to the treatment effect of sows fed a fibrous diet. Feeding high-fiber diets for sows may indirectly affect the weaning weight of piglets by regulating milk secretion. A greater number of sows and litters are needed to explore whether feeding sows a high-fiber diet during gestation can improve colostrum production or composition in further research.

### 4.3. Serum Biochemical Indexes

Measuring serum biochemical parameters of farm animals can provide important information regarding health and metabolism. The change in serum PUN concentration can reflect the whole body status of protein and amino acid metabolism and utilization in animals, when the amino acid metabolism in the body is strong, the serum urea nitrogen concentration will decrease [40,41]. It is known that serum globulin is synthesized and secreted by the immune organs of the animal, which is closely related to the body’s immunity. Changes in total protein and albumin content can reflect the metabolism and absorption of proteins by the animal body, of which ALB is the most important indicator of protein metabolism [42]. The content of TP and globulin in the CM group and SCR groups was significantly lower than in other groups, and the PUN content was significantly increased. This result indicates that the addition of CM and SCR diets during gestation will reduce protein metabolism and reduce protein utilization efficiency, and it may reduce animal immunity. In addition, our experimental results showing a decreased PUN level and increased TP content in the CG group, indicating that CG diet increases nitrogen deposition in sows and is beneficial to the utilization of protein and amino acids in the diet. The different effects of different fiber sources on TP, PUN, and ALB in the experiment may be related to the different sources of dietary fiber affecting the nitrogen utilization efficiency of the organism.

When fiber reaches the hindgut, it is used by gut microbes as a substrate for fermentation [43]. The short-chain fatty acids produced by fermentation are absorbed into the blood through the intestinal epithelium, and acetic acid can enter the liver with the blood circulation to participate in the body’s metabolism; propionic acid participates in gluconeogenesis, and butyric acid is mainly directly used by intestinal epithelial cells [44]. When short-chain fatty acids are absorbed by the intestine, glucose absorption in the intestine is reduced, and glucose levels in the blood can be stabilized [45]. The body will maintain a stable blood glucose concentration for a long time after the animal consumes dietary fiber [46]. This shows that the body of the animal has a stronger ability to regulate GLU, which may be the reason why GLU supplemented with fiber feed during gestation and lactation has not changed.

Serum AST and ALT have been proposed as indicators of depressed liver function. The increased activity of AST and ALT in serum suggests liver cell damage and leaching of these enzymes into the blood [47], and an excessive accumulation of these enzymes in the serum often prefigures liver injury [48,49]. Under normal conditions, AST and ALT levels are maintained at their highest levels in cardiomyocytes and liver, while blood levels are low. Once the myocardium and liver of the body are damaged, a large amount of AST and ALT in the liver are released into the blood, resulting in an increase in the activity of both enzymes in this compartment [50]. In this study, no changes in ALT and AST levels were observed in the group supplemented with the five high-fiber diets during gestation and lactation. We speculate that supplementation with fiber feed during gestation will not cause free radical damage to the liver.

### 4.4. Composition of Colostrum and Milk

The quality of milk is mainly reflected in the growth performance of piglets. Colostrum can increase piglet immunity and can also provide piglets with a variety of other nutrients, which is beneficial for their development. Thus, it is very important for piglets to consume more milk as early as possible. The development of the mammary gland during gestation directly affects milk production during lactation. Some research results have shown that high energy intake affects the development of sow mammary glands; in particular, increased intake of energy after 75 days of gestation is not conducive to mammary gland development [51]. Therefore, high-fiber diets supplementation during late gestation will not affect colostrum production, as there is a sufficient energy reserve in the sow to satisfy milk production. Studies have found that reducing energy intake during gestation can increase the fat content of colostrum [52]. The addition of fiber to the diet can reduce the energy concentration in the feed and avoid the high energy intake that affects milk secretion. In the current study, no significant differences were observed in the colostrum and milk fat, lactose, protein, and total solid content in the six treatment groups. However, colostrum milk fat showed a tendency to increase compared with that in the control group. Kirchgessner et al. [53] found that high-fiber diets promote milk fat content in sow colostrum and regular milk. However, some studies have shown that a high-fiber diet increases milk production compared with the control group, with no significant difference in milk composition [2,54], milk protein content generally is not affected by diet [55]. It was also reported that dietary fiber level during gestation did not affect total solids and lactose content of colostrum or milk [56]. These inconsistent results may be explained by different sources of fiber feed and different physical and chemical properties. The effects of different sources of fiber on sow milk composition have been less reported, necessitating further experiments for confirmation.

### 4.5. Diet Nutrient Digestibility

There are many reports on the effects of dietary fiber on animal nutrient digestibility, but most studies have suggested that dietary fiber reduces dietary nutrient digestibility. In this study, supplementation with CG diet significantly increased the ATTD of the NDF and ADF compared with those in the other groups. The ATTD of CP, EE, Ca, and P of RBM, CM, and WB groups were significantly reduced compared with those in the control group. Dietary fiber can promote the flow of dry matter and reduce energy utilization in the ileum and feces and the digestibility of starch, crude protein, lipids, and energy [57,58,59], With increases in dietary fiber of diet, the apparent digestibility of crude protein, crude fat and energy in feces is reduced [60]. Many studies have shown that dietary fiber reduces the residence time of chyme in the digestive tract and increases the speed of circulation, thereby reducing the digestibility of almost all nutrients [61,62].

The research by Girard et al. [63] showed that sow utilization of calcium, phosphorus, copper, and zinc decreased after feeding with a high-fiber diets (wheat bran/corn cob). Due to the strong cation exchange capacity of dietary fiber, it can absorb mineral elements. Some people think that this is the main reason for the reduced utilization of mineral elements. In addition, when chyme passes through the small intestine, the hydration of soluble fiber and water increases its viscosity, which promotes the formation of an immobile water layer, hinders contact between the chyme and digestive enzymes, and reduces the digestibility of nutrients [64]. In addition, no differences in the ATTD of the nutrients were observed between treatments during lactation because the lactating sows were fed the same diet. The results of this experiment revealed no negative effects on nutrient digestibility in gestating sows fed diets supplemented with CG and SCR, which represent a good source of fiber in sow feed.

## 5. Conclusions

This study has demonstrated that compared with the 8.43% NDF level in the control group, with the exception of the WB diet, there were no significant differences in sow performance in the experimental groups increased the NDF level to 16%. The use of CM, WB, CG, and RBM as a fiber source in feed had the potential to improve the average weaning weight of piglets. Furthermore, sows supplemented with CG diet had a lower PUN and a higher TP plasma concentration on day 100 of gestation, and the digestibility of NDF and ADF was higher in the CG diet.

## Figures and Tables

**Table 1 animals-11-00415-t001:** Ingredients and nutrient levels in the gestation and lactation diets.

Items	Gestation ^1^						Lactation
	CON	WB	CM	SCR	CG	RBM	
Ingredients (g/kg of diet)							
Corn	791.5	499.5	562.5	672.6	610.5	419.5	502
Soybean meal (46%)	170	99	166	112	79	79	76
Wheat bran	-	364	-	-	-	-	8
Chicory meal	-	-	236	-	-	-	-
Soybean curd residue	-	-	-	178	-	-	-
Corn gluten	-	-	-	-	270	-	-
Rice bran meal	-	-	-	-	-	465	-
Fish meal	-	-	-	-	-	-	25
Wheat germ	-	-	-	-	-	-	100
Soybean oil	-	-	-	-	-	-	10
Paddy	-	-	-	-	-	-	80
Extruded soy flour	-	-	-	-	-	-	90
Limestone	10	11	10	9	10	10	15
Dicalcium phosphate	13	11	13	13	13	11	11
Choline chloride (50%)	3.5	3.5	3.5	3.5	3.5	3.5	2
Lysine (98%)	-	-	1	-	2	-	-
Salt	4	4	4	4	4	4	4
Vitamin-mineral premix ^a^	8	8	8	8	8	8	9
Chemical composition (%)							
Crude protein ^b^	14	14	14	14	14	14	16.5
NDF ^b^	8.43	16	16	16	16	16	11.9
CF	2.06	4.09	7.90	3.26	4.93	5.72	1.85
Total phosphors ^b^	0.55	0.75	0.52	0.58	0.62	1.17	0.72
Calcium ^b^	0.75	0.75	0.75	0.75	0.75	0.75	1.00
Sodium ^b^	0.25	0.25	0.25	0.25	0.25	0.25	0.27
Chloride ^b^	0.40	0.40	0.40	0.40	0.40	0.40	0.40
Available phosphorus ^c^	0.35	0.35	0.35	0.35	0.35	0.35	0.40
Metabolizable Energy (MJ/kg) ^c^	13.81	12.00	13.41	13.52	12.49	12.55	13.62
Total lysine ^c^	0.72	0.73	0.84	0.74	0.71	0.77	1.06
Availablelysine ^c^	0.60	0.60	0.60	0.60	0.60	0.60	0.90

Abbreviations: NDF, neutral detergent fiber, CF, crude fiber. ^1^ CON, control; WB, wheat bran; CM, chicory meal; SCR, soybean curd residue; CG, corn gluten; RBM, rice bran meal. ^a^ Provided the following (per kg of diet): vitamin A, 12,642.67 IU; vitamin D_3_, 1966.64 IU; vitamin E, 44.95 mg; vitamin K_3_, 4.42 mg; vitamin B1, 4.05 mg; vitamin B_2_, 8.99 mg; vitamin B_6_, 5.46 mg; vitamin B_12_, 0.04 mg; pantothenic acid, 25.29 mg; nicotinic acid, 33.38 mg; folic acid, 1.6 mg; biotin, 0.22 mg; choline, 1312.5 mg; 0.2 mg of Co as CoCl_3_.6H_2_O; 0.5 mg of I as Ca(IO_3_)_2_; 0.4 mg of Se as Na_2_SeO_3_·H_2_O; 69, 53 or 164 mg of Mn as MnSO_4_; 140, 143 or 179 mg of Zn as ZnSO_4_; 166, 251 or 240 mg of Fe as FeSO_4_; 25, 26 or 26 mg of Cu as CuSO_4_. ^b^ Analyzed values. ^c^ Calculated chemical concentrations using values for feed ingredients from the NRC (2012).

**Table 2 animals-11-00415-t002:** Composition analysis of fiber sources in the diet.

Items	Treatments ^1^				
	WB	CM	SCR	CG	RBM
Chemical composition (%)					
NDF	29.71	40.2	30	37.52	25.13
CP	16	8.2	20	18.3	15
Ca	0.14	0.62	0.33	0.15	0.22
P	0.99	0.09	0.62	0.7	1.77

Abbreviations: NDF, neutral detergent fiber, CP, crude protein, Ca, calcium, P, phosphorus. ^1^ WB, wheat bran; CM, chicory meal; SCR, soybean curd residue; CG, corn gluten; RBM, rice bran meal.

**Table 3 animals-11-00415-t003:** Effects of different fiber sources on reproductive performance and body condition of sows during gestation and lactation.

Items	Treatments ^1^						*p*-Value
	CON	WB	CM	SCR	CG	RBM	
Sow BW (kg)							
Day 1 of gestation	225.83 ± 14.16	224.50 ± 30.08	222.67 ± 16.18	218.00 ± 19.23	237.20 ± 37.54	219.40 ± 24.50	0.478
Day 107 of gestation	289.50 ± 11.27 ^ab^	268.82 ± 20.70 ^b^	289.50 ± 17.50 ^ab^	281.17 ± 19.99 ^b^	305.44 ± 31.87 ^a^	277.70 ± 28.15 ^b^	0.009
Day 21 of weaning	257.33 ± 11.59 ^ab^	238.09 ± 31.42 ^c^	258.67 ± 19.87 ^ab^	247.50 ± 12.42 ^ab^	274.55 ± 30.89 ^a^	248.50 ± 28.95 ^ab^	0.045
Sow gestation BW change (kg)	63.37 ± 15.46 ^ab^	44.32 ± 15.03 ^c^	66.83 ± 13.13 ^ab^	63.17 ± 15.53 ^ab^	68.24 ± 14.57 ^a^	58.30 ± 11.64 ^bc^	0.021
Sow lactation BW change (kg)	32.18 ± 6.07	30.73 ± 5.03	30.83 ± 5.00	33.67 ± 3.56	30.89 ± 6.15	29.20 ± 5.60	0.147
Sow back fat (mm)							
Day 1 of gestation	17.40 ± 2.37	16.23 ± 2.63	17.37 ± 2.32	17.20 ± 2.48	17.09 ± 2.78	16.50 ± 2.99	0.849
Day 107 of gestation	20.90 ± 1.76 ^a^	18.23 ± 2.61 ^b^	20.83 ± 1.03 ^a^	20.03 ± 0.96 ^ab^	20.96 ± 1.77 ^a^	19.45 ± 1.84 ^b^	0.016
Day 21 of weaning	18.14 ± 1.22	16.27 ± 2.25	18.10 ± 1.16	17.98 ± 1.19	18.05 ± 2.00	17.20 ± 2.20	0.144
Sow back fat gain in gestation (mm)	3.49 ± 1.41 ^ab^	1.94 ± 0.59 ^c^	3.47 ± 1.03 ^ab^	2.83 ± 0.79 ^ab^	3.87 ± 1.56 ^a^	2.95 ± 0.96 ^ab^	0.037
Sow back fat loss on lactation (mm)	2.76 ± 1.02	1.97 ± 0.89	2.73 ± 1.13	2.04 ± 0.91	2.90 ± 1.33	2.25 ± 1.05	0.106

Abbreviations: BW, body weight. ^1^ CON, control; WB, wheat bran; CM, chicory meal; SCR, soybean curd residue; CG, corn gluten; RBM, rice bran meal. Note: The values are given as the mean ± SD (*n* = 12). ^a,b,c^ Mean values within a row without a common superscript differ significantly (*p* < 0.05).

**Table 4 animals-11-00415-t004:** Effects of different fiber sources on piglet performance.

Items	Treatments ^1^						*p*-Value
	CON	WB	CM	SCR	CG	RBM	
Number of total piglets born/litter	11.33 ± 0.82	11.87 ± 1.5	10.75 ± 0.50	11.20 ± 1.48	12.10 ± 1.10	11.50 ± 1.05	0.410
Number of piglets born alive/litter	11.17 ± 0.75	10.38 ± 1.85	9.83 ± 1.30	10.80 ± 1.10	11.45 ± 1.86	11.40 ± 1.26	0.244
Number of piglets at weaning/litter	10.00 ± 1.00	9.63 ± 1.85	8.71 ± 0.95	9.83 ± 0.75	10.18 ± 2.08	9.60 ± 1.84	0.566
Survival rate at birth (%)	0.99 ± 0.02	0.93 ± 0.08	0.98 ± 0.08	0.96 ± 0.09	0.91 ± 0.11	0.95 ± 0.09	0.343
Survival rate at weaning (%)	0.90 ± 0.08	0.93 ± 0.06	0.92 ± 0.11	0.94 ± 0.11	0.95 ± 0.07	0.97 ± 0.06	0.673
Litter birth weight (kg)	18.19 ± 1.95	18.24 ± 2.71	16.52 ± 3.44	17.78 ± 2.81	18.78 ± 2.56	19.14 ± 3.31.	0.369
Litter weight at weaning (kg)	59.73 ± 5.93	65.17 ± 6.34	62.23 ± 7.15	62.69 ± 6.92	68.75 ± 5.08	62.28 ± 10.21	0.836
Average birth weight (kg)	1.58 ± 0.24	1.54 ± 0.11	1.69 ± 0.27	1.57 ± 0.23	1.52 ± 0.19	1.64 ± 0.26	0.502
Average weaning weight (kg)	5.85 ± 1.11 ^b^	6.71 ± 0.66 ^a^	6.80 ± 1.04 ^a^	6.22 ± 0.40 ^ab^	6.76 ± 0.66 ^a^	6.64 ± 0.57 ^a^	0.035
Piglets day 0–21 ADG (kg/d)	0.21 ± 0.06	0.25 ± 0.03	0.24 ± 0.05	0.22 ± 0.02	0. 25 ± 0.03	0.23 ± 0.02	0.160

Abbreviations: ADG, average daily gain. ^1^ CON, control; WB, wheat bran; CM, chicory meal; SCR, soybean curd residue; CG, corn gluten; RBM, rice bran meal. Note: The values are given as the mean ± SD (*n* = 12). ^a,b^ Mean values within a row without a common superscript differ significantly (*p* < 0.05).

**Table 5 animals-11-00415-t005:** Effect of different fiber sources on plasma biochemical parameters during gestation and lactation.

Items	Treatments ^1^						*p*-Value
	CON	WB	CM	SCR	CG	RBM	
Day 100 of gestation							
GLU (mmol/L)	5.17 ± 0.70	4.89 ± 0.40	4.57 ± 0.48	5.13 ± 0.54	4.71 ± 0.72	4.62 ± 0.41	0.148
PUN (mmol/L)	5.04 ± 0.72 ^b^	5.15 ± 0.62 ^ab^	5.82 ± 0.87 ^a^	5.12 ± 0.66 ^ab^	4.16 ± 0.51 ^c^	4.81 ± 0.58 ^bc^	0.001
TG (mmol/L)	0.81 ± 0.34	0.78 ± 0.14	0.80 ± 0.29	0.82 ± 0.35	0.74 ± 0.13	0.72 ± 0.13	0.952
ALT (IU/L)	50.48 ± 6.09	59.28 ± 5.75	47.36 ± 8.35	47.86 ± 6.85	45.09 ± 5.55	46.51 ± 5.76	0.111
AST(IU/L)	24.78 ± 4.08	25.94 ± 2.47	31.68 ± 6.77	24.25 ± 6.46	25.04 ± 2.36	25.39 ± 2.20	0.492
AKP (IU/L)	44.52 ± 4.17	50.66 ± 6.58	52.32 ± 7.40	48.06 ± 4.54	54.15 ± 3.77	56.55 ± 3.96	0.262
TP (g/L)	77.93 ± 3.83 ^b^	81.91 ± 5.26 ^ab^	72.94 ± 2.34 ^c^	73.19 ± 3.27 ^c^	82.86 ± 5.56 ^a^	80.63 ± 5.09 ^ab^	<0.001
ALB (g/L)	43.76 ± 3.92	44.24 ± 44.69	43.75 ± 1.70	44.10 ± 2.70	42.88 ± 2.18	42.91 ± 2.32	0.953
Globulin(g/L)	34.17 ± 4.46 ^b^	37.67 ± 5.31 ^a^	29.19 ± 3.12 ^c^	29.09 ± 2.97 ^c^	39.98 ± 3.46 ^ab^	37.71 ± 3.53 ^ab^	<0.001
Day 14 of lactation							
GLU (mmol/L)	4.79 ± 0.57	5.08 ± 0.75	5.16 ± 0.81	5.48 ± 0.71	5.60 ± 0.76	5.19 ± 0.70	0.442
PUN (mmol/L)	5.50 ± 1.03	6.36 ± 0.86	6.21 ± 1.47	6.23 ± 1.01	5.50 ± 0.88	5.80 ± 0.99	0.396
TG (mmol/L)	0.27 ± 0.07	0.30 ± 0.07	0.22 ± 0.06	0.23 ± 0.08	0.32 ± 0.08	0.26 ± 0.09	0.353
ALT (IU/L)	50.98 ± 4.59	45.48 ± 6.20	50.16 ± 6.78	53.24 ± 6.23	48.73 ± 6.57	48.16 ± 4.18	0.754
AST(IU/L)	24.4 ± 2.84	25.70 ± 6.10	32.83 ± 5.19	27.39 ± 5.12	28.36 ± 4.93	32.26 ± 5.21	0.315
AKP (IU/L)	65.53 ± 6.96	62.09 ± 5.77	61.15 ± 6.51	78.63 ± 6.63	63.99 ± 6.52	57.67 ± 5.32	0.910
TP (g/L)	77.67 ± 4.94 ^abc^	79.78 ± 2.90 ^ab^	75.78 ± 3.51 ^bc^	74.82 ± 3.22 ^c^	80.77 ± 3.60 ^a^	78.63 ± 5.47 ^abc^	0.041
ALB(g/L)	45.04 ± 2.49	44.48 ± 3.18	46.35 ± 2.28	45.20 ± 1.42	44.71 ± 2.47	44.35 ± 2.67	0.634
Globulin (g/L)	32.63 ± 3.52 ^ab^	35.30 ± 4.51 ^a^	29.43 ± 2.07 ^b^	29.62 ± 2.49 ^b^	36.06 ± 4.87 ^a^	34.28 ± 3.72 ^a^	0.011

Abbreviations: GLU, glucose; PUN, plasma urea nitrogen; TG, glycerin trilaurate; ALT, alanine aminotransferase; AST, aspartate aminotransferase; AKP, alkaline phosphatase; TP, total protein; ALB, albumin; GLB, globulin. ^1^ CON, control; WB, wheat bran; CM, chicory meal; SCR, soybean curd residue; CG, corn gluten; RBM, rice bran meal. Note: The values are given as the mean ± SD (*n* = 12). ^a,b,c^ Mean values within a row without a common superscript differ significantly (*p* < 0.05).

**Table 6 animals-11-00415-t006:** Effect of different fiber sources on the composition of colostrum and milk during gestation and lactation.

Items	Treatments ^1^						*p*-Value
	CON	WB	CM	SCR	CG	RBM	
Colostrum							
Fat (%)	3.57 ± 1.14	3.63 ± 1.05	3.77± 0.92	4.03 ± 0.93	4.37 ± 0.89	3.87 ± 0.78	0.576
Lactose (%)	3.36 ± 0.42	3.35 ± 1.18	3.34 ± 0.63	3.27 ± 0.67	4.01 ± 0.73	3.20 ± 0.91	0.551
Protein (%)	17.78 ± 2.36	18.64 ± 2.69	19.55 ± 2.09	18.65 ± 3.55	16.39 ± 3.34	16.80 ± 1.14	0.303
Total solid (%)	24.44 ± 2.47	25.53 ± 3.55	26.05 ± 2.08	24.65 ± 3.61	23.85 ± 2.12	23.61 ± 2.49	0.583
Milk (day 14 of lactation)							
Fat (%)	7.37 ± 0.84	6.71 ± 0.85	7.73 ± 0.64	6.71 ± 1.04	7.54 ± 0.84	7.32 ± 1.22	0.303
Lactose (%)	6.30 ± 0.32	6.06 ± 0.28	5.90 ± 0.42	5.70 ± 1.06	5.85 ± 0.70	6.00 ± 0.92	0.515
Protein (%)	5.08 ± 0.37	4.70 ± 0.55	5.18 ± 0.71	4.62 ± 0.86	5.36 ± 0.62	4.96 ± 0.41	0.289
Total solid (%)	19.77 ± 2.83	18.50 ± 2.66	19.97 ± 2.97	18.39 ± 2.90	19.62 ± 1.47	19.18 ± 3.22	0.279

^1^ CON, control, WB, wheat bran, CM, chicory meal, SCR, soybean curd residue, CG, corn gluten, RBM, rice bran meal. Note: The values are given as the mean ± SD (*n* = 12).

**Table 7 animals-11-00415-t007:** Effect of different fiber sources on the apparent total tract digestibility of nutrients during gestation.

Items	Treatments ^1^						*p*-Value
	CON	WB	CM	SCR	CG	RBM	
Apparent digestibility (%)							
NDF	59.43 ± 4.09 ^b^	58.92 ± 4.89 ^b^	48.47 ± 2.36 ^c^	59.58 ± 4.07 ^b^	71.32 ± 4.15 ^a^	48.46 ± 3.27 ^c^	<0.001
ADF	49.03 ± 4.49 ^b^	52.70 ± 4.12 ^b^	37.56 ± 3.27 ^c^	49.26 ± 4.05 ^b^	67.69 ± 5.16 ^a^	51.88 ± 4.85 ^b^	<0.001
CP	89.13 ± 2.01 ^ab^	86.18 ± 2.81 ^c^	83.88 ± 1.36 ^d^	89.50 ± 2.08 ^a^	87.76 ± 2.93 ^b^	81.60 ± 2.34 ^e^	<0.001
EE	84.97 ± 5.54 ^a^	69.14 ± 5.99 ^c^	42.07 ± 4.12 ^d^	87.71 ± 6.33 ^a^	77.55 ± 5.93 ^b^	66.43 ± 5.05 ^c^	<0.001
Ca	66.70 ± 5.52 ^a^	41.56 ± 6.67 ^b^	38.27 ± 4.17 ^bc^	67.89 ± 5.37 ^a^	43.06 ± 7.05 ^b^	34.17 ± 3.64 ^d^	<0.001
P	66.36 ± 6.63 ^a^	40.69 ± 5.63 ^b^	39.43 ± 4.85 ^b^	67.65 ± 7.38 ^a^	44.46 ± 8.50 ^b^	40.93 ± 6.46 ^b^	<0.001
CF	57.54 ± 4.37 ^b^	56.45 ± 4.16 ^b^	52.31 ± 4.31 ^c^	58.35 ± 5.09 ^b^	62.25 ± 5.11 ^a^	51.36 ± 3.69 ^c^	<0.001

Abbreviations: NDF, neutral detergent fiber, ADF, acid detergent fiber, CP, crude protein, EE, crude fat, Ca, calcium, P, phosphorus, CF, crude fiber. ^1^ CON, control, WB, wheat bran, CM, chicory meal, SCR, soybean curd residue, CG, corn gluten, RBM, rice bran meal. Note: The values are given as the mean ± SD (*n* = 12). ^a,b,c,d,e^ Mean values within a row without a common superscript differ significantly (*p* < 0.05).

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
