# Peer review of "The Effect of Increasing Neutral Detergent Fiber Level through Different Fiber Feed Ingredients throughout the Gestation of Sows"

_animals, 2021, doi:10.3390/ani11020415_

Round 1

Reviewer 1 Report

Shi et al. have submitted a manuscript entitled “The effect of increasing neutral detergent fiber level through different fiber feed ingredients throughout the gestation of sows” for publication in Animals.

The authors have selected 5 common household and production wastes as fiber sources and added them to the diet of pregnant sows to explore the impact on sows and piglets. They analyzed reproductive performance, plasma biochemistry parameters, colostrum and milk composition, and nutrient digestibility. The authors observed that adding corn gluten feed could significantly improve digestibility and weaning of piglets, body weight, and other parameters.

The following points are recommended for strengthening of the manuscript.

Major:

1) The authors might address more specifically to the single differences in diet, e.g. limestone, soybean meal, and other compounds which differ. Have there been control experiments to exclude possible effects?

2) The authors might include SEM to presented single values for more transparency.

3) n number of litters appears not to allow a convincing interpretation of data. The authors might be more careful in this respect, or extend the study.

Minor:

  1. Simple summary could be more exact and detailed, referring to results in past tense.
  2. All Abbreviations should be written in full when mentioned for the first time.
  3. Generally, orthography needs to be checked.

Reviewer 2 Report

The authors present a manuscript that investigates the effects of different fiber feed diets on sows biochemical, weight, fat and milk quality parameters, among other tests. The manuscript seem to have a sound experimental and scientific design. The manuscript is interesting, has relevant findings and investigated many aspects of sows quality, including milk which resulted in a work that has a great bulk of information.

Since the work aims to explore differences in parameters of different variables, that is the different diets, it is paramount that the statistical design is clearly presented, as it is the most important factor in the discussion. In my opinion, the work can still improve in their statistical design and presentation. It was unclear to me, how the Anova test was performed. The authors should clearly present what were their null and consequently alternative hypothesis. The null hypothesis probability values can be used only to ascertain whether there is a group with significant difference from the other (but which group is different is unknown), but paired tests such as tukey, should show which parameter is different from which. The authors display that they performed the tukey test in their methodology but the discussion is performed on the p values for the anova test which does not display accurately what the authors are discussing.

I suggest that the authors perform the Anova test, Tukey test and a variance test (such as Levenne) using the control average values as your null hypothesis; in this way you will be able to accurately see if the treatments are yielding different results from the control. Moreover, differences in variances in the parameters can generate new insights into the diets effects. Clearly address your statistical design and display which group is statistically different from the control with the tukey values.

Table 1 has confusing unit labelling, it is suggested that either the authors change the values to the same unit or to separate the in subsections that display a singular unit in each subsection

L182 – The null hypothesis is not well explained here. Clearly explain what was the null hypothesis of the test

Table 2 “a,b,cMeans within a row without a common letter indicate a significant difference (p < 0.05)” This sentence is very confusing.

The probability of the null hyphotesis greater than the significance level tested (0.05) indicates that the variables have different means, considering your null hypothesis was of equality between variables, however, which feeding treatments were different and which were not? Where is the paired tukey test?

The authors should display their values with standard deviation SD instead of SEM, for a clearer understanding

Round 2

Reviewer 1 Report

--

Reviewer 2 Report

The suggestions have been satisfactory attended to and have been adressed properly. I believe the manuscript is suitable for publication.